# Photoinduced monooxygenation involving NAD(P)H-FAD sequential single-electron transfer

Simon Ernst[1], Stefano Rovida[2], Andrea Mattevi [2], Susanne Fetzner [1] & Steffen L. Drees [1✉]

Light-dependent or light-stimulated catalysis provides a multitude of perspectives for implementation in technological or biomedical applications. Despite substantial progress made in the field of photobiocatalysis, the number of usable light-responsive enzymes is still very limited. Flavoproteins have exceptional potential for photocatalytic applications because the name-giving cofactor intrinsically features light-dependent reactivity, undergoing photoreduction with a variety of organic electron donors. However, in the vast majority of these enzymes, photoreactivity of the enzyme-bound flavin is limited or even suppressed. Here, we present a flavoprotein monooxygenase in which catalytic activity is controllable by blue light illumination. The reaction depends on the presence of nicotinamide nucleotide-type electron donors, which do not support the reaction in the absence of light. Employing various experimental approaches, we demonstrate that catalysis depends on a protein-mediated photoreduction of the flavin cofactor, which proceeds via a radical mechanism and a transient semiquinone intermediate.

---

[1] Institute for Molecular Microbiology and Biotechnology, University of Münster, Corrensstr. 3, 48149 Münster, Germany. [2] Department of Biology and Biotechnology "Lazzaro Spallanzani", University of Pavia, via Ferrata 9, 27100 Pavia, Italy. ✉email: s.drees@uni-muenster.de

Photoswitchable catalysts hold great promise for a variety of applications. It is therefore not surprising that substantial efforts are made for the development of artificial photochemical catalysts, photoenzymes[1], and synthetic optogenetic tools[2]. Based on the photochemical properties of flavin, flavoproteins bear particularly great potential for optoenzymatic applications. For example, naturally occurring flavin cofactors such as FMN and FAD are known to undergo photoinduced reduction[3]. In the process, various electron donors, such as amino acids, aromatic compounds, EDTA, but also intramolecular substituents are oxidized[4]. However, so far, only a handful of flavin photoproteins have been identified. Amongst them, the ubiquitous and well-characterized group of DNA photolyases is difficult to exploit for biotechnological applications due to their sophisticated adaptations to the DNA substrate. Another heterogeneous class of flavin photoproteins comprising LOV domains, BLUF proteins, and cryptochromes employ the photoactivated flavin redox cycle for signaling purposes without the involvement of an external organic substrate. While major breakthroughs were made in the photochemical use of flavin ene-reductases[5–7] and the recently discovered photodecarboxylase[8,9], the applicability of most flavoproteins is impeded due to efficient shielding of protein-bound cofactors against light-induced excitation[10]. Most likely, the prevention of photoinduced generation of reactive oxygen species in the cellular environment, usually achieved by efficient quenching of singlet or triplet excited states[11], has been a strong selective pressure in the evolution of these enzymes. It is therefore generally accepted that photoreduction, when observed in flavoenzymes, can be attributed to photoreactivity of free flavin, which is present in small quantities in most flavoprotein preparations due to degradation and/or binding equilibria[10]. Undirected, diffusive charge transfer subsequently leads to the observed reduction of the protein-internal flavin cofactor. This mechanism has been extensively tested for biotechnological application with various classes of non-photochemically accessible flavoenzymes[12].

In this study, we show that the catalytic activity of the flavoprotein monooxygenase (FPMO) PqsL can be triggered by blue-light illumination. The enzyme is involved in the biosynthesis of 2-alkyl-4-hydroxyquinoline N-oxides (AQNO) in the opportunistic pathogen Pseudomonas aeruginosa. It is homologous to group A FPMOs[13] and has a relatively high similarity to p-hydroxybenzoate hydroxylase (pHBH), the most thoroughly studied FPMO[14]. Group A monooxygenases usually hydroxylate mono- or polysubstituted aromatic rings and utilize NAD(P)H as hydride donor. In contrast, PqsL catalyzes the hydroxylation of an aromatic-amine group, a rather uncommon reaction, and remarkably shows no intrinsic catalytic activity with NAD(P)H as cosubstrate[15]. Here, we provide experimental evidence that the protein-internal FAD of PqsL is reduced by NAD(P)H in a light-dependent, enzyme-associated reaction which is severalfold faster than the photoreduction of free flavin. Structural and functional studies indicate that canonical transient binding of NAD(P)H is compatible with photoexcitation.

Using an engineered variant of PqsL, we further demonstrate that stabilization of the protein-cofactor complex does not diminish photoreactivity, indicating that primarily the conformation of the flavin ring and its microenvironment are relevant for photoreduction. We moreover elucidate the charge transfer between NAD(P)H and FAD, in which the protein-bound flavin reacts similarly to free flavin, involving a two-step reaction with single-electron transfer, leading to a transient neutral semiquinone intermediate.

## Results

**Blue light induces catalytic activity of PqsL.** PqsL is a monooxygenase involved in AQNO biosynthesis in P. aeruginosa. Its catalytic activity was found to depend on the presence of reduced flavin, e.g., provided by a flavin reductase[15]. In that regard, the enzyme functions similarly to the two-component monooxygenases, in which the FAD cofactor migrates between a monooxygenase and a cognate reductase. In the process of analyzing PqsL cofactor exchange and charge transfer we discovered an interesting anomaly: the enzyme slowly oxidized NAD(P)H when the reaction was monitored in a spectrophotometer cuvette, whereas no consumption of the cosubstrate was observable when the reaction was analyzed by, e.g., HPLC. We concluded that the PqsL reductive half reaction must be positively influenced by light.

To test our hypothesis, we analyzed if PqsL-catalyzed product formation could be observed with NAD(P)H as electron donor depending on whether the samples were illuminated or not. Due to instability of the PqsL reaction product 2-hydroxylaminobenzoylacetate (2-HABA), we used a coupled assay for detection in which the substrate, 2-aminobenzoylacetate (2-ABA), and 2-HABA are enzymatically converted to the stable alkyl quinolone derivatives 2-heptyl-4(1H)-quinolone (HHQ) and 2-heptyl-4-hydroxyquinoline-N-oxide (HQNO), respectively (Fig. 1a). Both compounds can then be quantified by HPLC. As displayed in Fig. 1b, HQNO formation increased as linear function of blue-light intensity whereas no product was detected in the dark, indicating that NAD(P)H is not accepted as electron donor in the absence of light. We also grew cultures of P. aeruginosa PAO1 with blue-light illumination under aerobic conditions. However, no difference in AQNO/non-AQNO metabolite ratio was observed. Instead a complex phenotype with reduced general levels of AQ pathway metabolites and other secondary metabolites such as phenazines was observed in response to illumination, which is consistent with previous reports[16].

The photochemical reduction of flavins is an extensively studied process[10] and has been exploited for various biotechnological applications. To this end, a variety of efficient, cost-effective electron donors are used to drive photoreduction[17]. Expecting that the reduction of PqsL was merely a side reaction driven by residual free FAD in solution, we tested a set of typical sacrificial electron donors such as EDTA, L-glycine, triethylamine or ascorbic acid on their effect on the reaction. To our surprise, only ascorbic acid turned out to support the reaction, though not to the extent of NADPH. EDTA, L-cysteine, L-glycine, and triethylamine led to the generation of only trace amounts of the product (Fig. 1c). Furthermore, we did not observe any product formation when PqsL was substituted with free FAD. These results indicated that the PqsL reaction does not rely on the typical photochemistry of free flavin but is a strictly protein-associated process. Accordingly, NADPH is the preferred electron-donating substrate for the photoactivated enzyme.

In a further approach, we tested reduced 1-methylnicotinamide (MNAH), an electron donor known for its capability to support various flavin dependent redox enzymes by NAD(P)H-analogous hydride transfer to both, protein-bound and free flavin cofactors[18]. MNAH supported the PqsL reaction in the dark but showed no increase in efficiency when samples were illuminated. Since MNAH reduces FAD in solution at a faster rate than PqsL ($k_{FAD} = 3 \times 10^{-3} \, s^{-1} \, mM^{-1}$, $k_{PqsL} = 1 \times 10^{-3} \, s^{-1}$ $mM^{-1}$, Supplementary Fig. 1), it is conceivable that the observed activity stems from the reduction of free FAD rather than from the protein-associated reaction prevailing with the nicotinamide nucleotides.

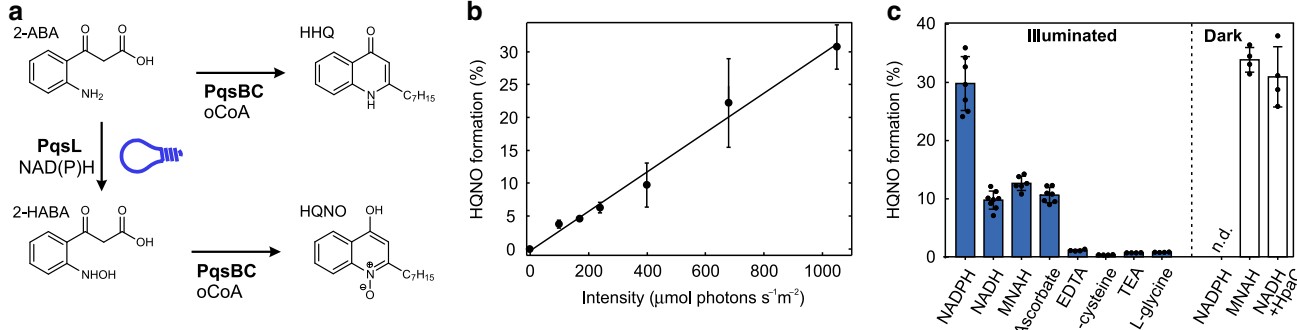

**Fig. 1 Blue-light-induced activity of PqsL. a** Reaction of PqsL within the alkyl quinolone biosynthetic pathway. The downstream/parallel reaction of PqsBC was used throughout the study for coupled assays to quantify PqsL catalytic activity. 2-HABA is unstable with a $t_{1/2}$ of 16 minutes. 2-ABA—2-aminobenzoylacetate; HHQ—2-heptyl-4(1H)-quinolone; 2-HABA—2-hydroxylaminobenzoylacetate; HQNO—2-heptyl-4-hydroxyquinoline-N-oxide; oCoA—octanoyl-CoA. **b** HQNO formation in the coupled enzyme assay as a function of light intensity (LED source, $\lambda_{max}$ 466 nm, 20 nm full width at half maximum, $n \geq 6$). **c** HQNO formation in the coupled enzyme assay depending on the PqsL electron donor ($n \geq 4$). The flavin reductase HpaC was used to generate reduced flavin in situ for PqsL light-independent activity with NADH. Error bars denote SD of independent experiments. MNAH—1-methyl-1,4-dihydronicotinamide, TEA—triethylamine.

Taken together, these results encouraged us to study the characteristics of PqsL photoreactivity in more detail. Due to the complex absorption characteristics of the PqsL-related organic compounds (2-ABA, 2-HABA, HHQ, and HQNO all interfere with NAD(P)H absorption in the UV-A range), the photoreductive reaction of PqsL was followed by monitoring light-induced oxygen consumption, caused either by unproductive reoxidation (uncoupling) or product formation.

Cyclic illumination of PqsL in the presence of NADPH (Fig. 2a) revealed that the reaction instantaneously responded to blue light and can therefore be triggered or stopped without temporal offset. The reaction velocity did not decrease over several cycles of illumination, indicating that the enzyme can perform multiple turnovers without oxidative damage. The reaction moreover shows a wavelength dependency consistent with the absorption spectrum of PqsL-bound FAD (Fig. 2b), clearly indicating that flavin excitation rather than excitation of NAD(P)H is triggering the photoreduction. Illumination with a simple blue LED is therefore sufficient to fully sustain the reaction.

Free flavins are known to be promiscuous regarding electron donors employed for photochemical reactions. For instance, besides electron transfer from EDTA or amino acids, flavin mononucleotide is prone to intramolecular photodegradation, with the flavin drawing electrons from the ribityl moiety[4]. To confirm that PqsL oxidizes the nicotinamide moiety of NAD(P)H, we analyzed illuminated samples of PqsL/NAD(P)H by HPLC. Figure 2c shows the time-dependent formation of $NAD^+$ from NADH as sole oxidation product of PqsL. Similar results were observed when NADPH was used instead of NADH, suggesting nicotinamide-specific oxidation of NAD(P)H by photosensitized PqsL. In contrast, little or no degradation products of FAD such as lumichrome are observed in response to illumination of PqsL (Supplementary Fig. 4).

Steady-state kinetics revealed typical Michaelis-Menten characteristics for the light-driven reaction of PqsL with NAD(P)H. Apparent $K_M$ values in the micromolar range ($157.5 \pm 18.9 \, \mu M$ for NADPH; $310.9 \pm 21.6 \, \mu M$ for NADH, errors given reflect SD calculated from regression confidence intervals) are indicative of specific enzymatic recognition of NAD(P)H (Fig. 2d). Compared to the photoreduction of free FAD, the enzymatic reduction reached higher overall reaction rates ($k_{cat} = 0.60 \pm 0.02 \, s^{-1}$). These may be limited by the lifetime of the protein-bound reduced flavin, which is stabilized to varying degrees by many monooxygenases and probably also by PqsL[19]. The most

significant relative increase of photoreduction was observed at $7.8 \, \mu M$ NADPH (PqsL $2.6 \times 10^{-2} \, s^{-1}$, FAD $0.0916 \times 10^{-2} \, s^{-1}$), where the enzymatic reduction was 30 times more efficient than the non-catalyzed reaction. This can be explained by the substantial amplification of the enzymatic reaction due to the formation of the Michaelis–Menten complex. In addition, due to the low degree of enzyme saturation at this concentration, the reduced complex has a relatively low abundance, so that the reaction is hardly affected by its slow decay. In agreement with the obtained product yields (Fig. 1c), PqsL showed a higher catalytic efficiency with NADPH than with NADH (Supplementary Table 1). Taken together, the steady-state kinetics provide evidence for an enzyme-mediated mechanism of FAD photoreduction and open up a perspective on how light-switchable enzymes based on PqsL could be implemented in, e.g., one-vessel cascaded biocatalytic reactions.

**pHBH fold with a characteristic substrate-binding site of PqsL.** Crystallization experiments with wild-type PqsL produced crystals of very high quality (Table 1). This allowed us to solve the three-dimensional structure of the enzyme bound to FAD and its substrate 2-ABA at 1.8 Å resolution. As expected, the overall conformation of PqsL is similar to that of p-hydroxybenzoate hydroxylase (pHBH; PDB entry 1PBE)[13,20–24] as indicated by a root-mean-square deviation of 2.1 Å for 322 Cα atoms (19% sequence identity). All the main features of the pHBH fold are conserved in PqsL. We therefore focused our analysis on the specific properties of PqsL's catalytic site where a molecule of 2-ABA could be located based on the electron density map. 2-ABA occupies a large cavity above the edge of the flavin ring. This binding mode and location are similar to that observed for the substrate of pHBH (Fig. 3a, Supplementary Fig. 5). Specifically, 2-ABA interacts with the main chain atoms of two active-site loops (residues 46–48 and 300–303) and its carboxylate group H-bonds to the backbone nitrogen of Asp47 and a water molecule. A critical observation was that the substrate aromatic ring is oriented with its amino group pointing away from the flavin. This binding orientation cannot sustain catalysis because the amino group must be in reach of the terminal oxygen of the flavin-hydroperoxide intermediate (Fig. 1a). To explain this conundrum, we performed steady-state kinetics at high substrate concentrations as in the crystallization experiments. Addition of the substrate led to a substantially decreased reaction velocity ($k_{cat} = 0.18 \pm 0.01 \, s^{-1}$)

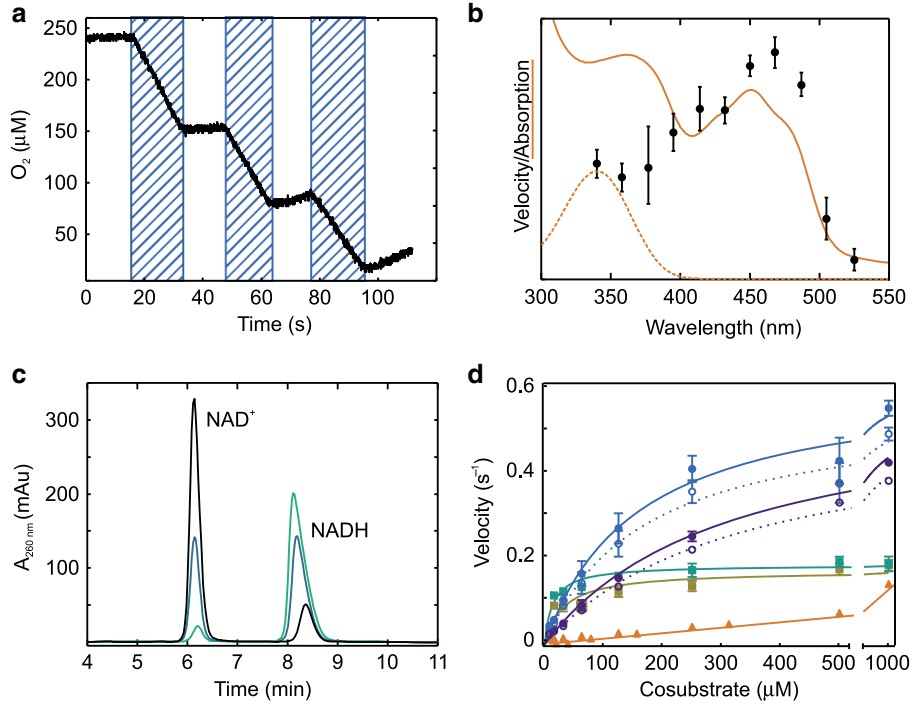

**Fig. 2 Characteristics of PqsL photoreactivity. a** PqsL photoreduction measured over several cycles of illumination (shaded intervals). The reaction of 10 μM PqsL was monitored by $O_2$ consumption, 1.25 mM NADPH was used as electron donor (one representative experiment shown). **b** Apparent velocity as a function of the wavelength of the illuminating light (black dots) together with the UV/Vis spectra of PqsL (orange, solid line) and NADH (orange, dashed line). **c** $NAD^+$ formation from NADH oxidation by PqsL after 1 min (light green), 5 min (petrol blue) and 10 min (black) of illumination, as monitored by HPLC (one representative experiment). **d** Kinetics of light-induced oxygen consumption by PqsL supplemented with NADPH (blue circles), NADH (violet circles), NADPH + 2-ABA (green squares) and NADH + 2-ABA (ocher squares) or free FAD + NADPH (orange triangles). Rates of the PqsL reactions were corrected for the contribution of residual free FAD (calculated from $K_D$ of the PqsL–FAD complex) and FAD photodegradation. Dashed lines/open circles show the rates of FAD-saturated PqsL prior to any corrections (i.e. rates as measured, normalized to FAD content). Line fits were calculated according to a Michaelis-Menten model. Parameters are provided in Supplementary Table 1. Experimental data on photoreduction of PqsL and free FAD with various alternative electron donors are provided in Supplementary Figs. 2 and 3. Error bars indicate SD of three identical measurements (panels **b**, **d**).

| Table 1 Crystallographic statistics. | | |
|---|---|---|
| | **Wild-type PqsL** | **PqsL-Q (R41Y, I43R, G45R, C105G)** |
| PDB code | 6SW2 | 6SW1 |
| Wavelength (Å) | 0.96600 | 0.96770 |
| Resolution range (Å) | 45.0–1.7 (1.74–1.70) | 38–1.8 (1.85–1.80) |
| Space group | $P2_12_12_1$ | $P2_12_12_1$ |
| Unit cell (Å) | 47.08 63.98 128.10 | 47.53 65.75 135.24 |
| Unique reflections | 40904 (2993) | 36923 (2735) |
| Multiplicity | 4.8 (4.8) | 3.2 (5.9) |
| Completeness (%) | 99.1 (98.8) | 96.8 (99.1) |
| Mean I/sigma (I) | 10.1 (1.0) | 17.3 (2.1) |
| R-merge (%) | 7.0 (120.7) | 4.1 (65.8) |
| $CC_{1/2}$ | 0.998 (0.511) | 0.999 (0.464) |
| R-work (%) | 17.5 | 19.0 |
| R-free (%) | 21.7 | 22.1 |
| Number of non-hydrogen atoms | 3196 | 3121 |
| Protein residues | 367[a] | 367[a] |
| RMS (bonds) (Å) | 0.012 | 0.014 |
| RMS (angles) (°) | 1.6 | 1.7 |
| Ramachandran favored (%) | 96.0 | 96.7 |
| Ramachandran allowed (%) | 3.7 | 2.8 |
| Ramachandran outliers (%) | 0.3 | 0.5 |

[a]The final models contain residues 3–370.

and a lower apparent $K_M$ (13.1 ± 2.2 μM) with NADPH (Fig. 2d). According to previous data, it is unlikely that amine hydroxylation is the rate-limiting step of the reaction[15]. Therefore, the observed effects most likely reflect an uncompetitive inhibition by 2-ABA, in which the oxidized and probably also the reduced PqsL–FAD complexes formally represent inhibited enzyme-cosubstrate species. Furthermore, substrate inhibition can also explain, why complete substrate conversion by PqsL has not been observed in any assay so far (a maximum of 50% conversion can be observed in a PqsL-PqsBC-coupled assay in oxygen-saturated buffer[15]).

Flavin CD spectroscopy showed that the wild-type enzyme shows a low ellipticity of the chromophore, corresponding to a high degree of conformational freedom, whereas addition of 2-ABA led to a more rigid conformation of the flavin, as suggested by an increased ellipticity with a slightly shifted spectrum (Fig. 4a). The crystallized 2-ABA-stabilized oxidized PqsL could therefore be the dominant enzyme-substrate-inhibitor complex, with a $K_D$ of 38.44 ± 3.98 μM (Supplementary Fig. 6). A simple 180° flipping rotation around the C1–C4 axis of the aromatic ring would position the amino group of the 2-ABA substrate toward the flavin, in the position expected for amine hydroxylation. In essence, the crystal structure of PqsL shows that the aromatic-amine hydroxylase activity of PqsL can be implemented into the pHBH fold without drastic alterations of the overall protein

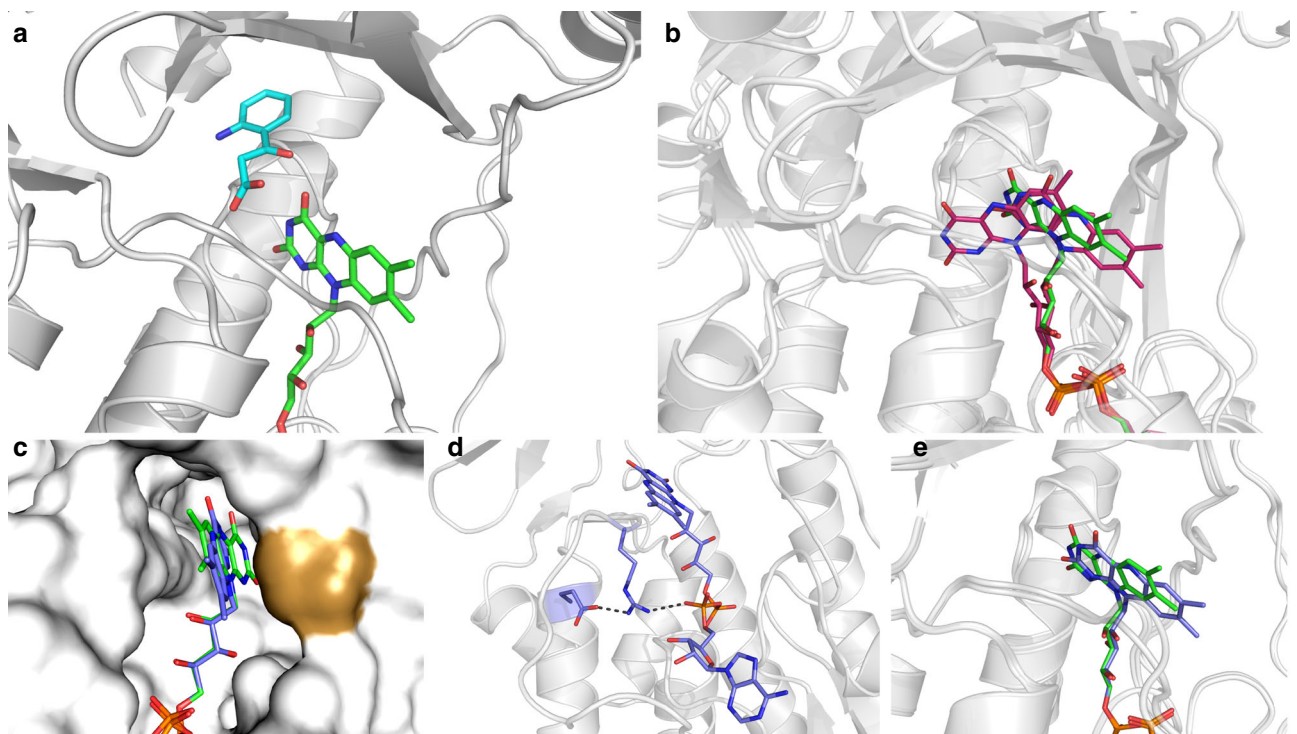

**Fig. 3 Crystallographic analysis of FAD and 2-ABA coordination in PqsL. a** Unproductive 2-ABA (cyan) binding in PqsL (PDB ID: 6SW1). **b** Comparison of the flavin conformation in PqsL (green) to the IN and OUT conformations of FAD in pHBH (red, PDB ID: 1PBB, 1PBE). **c** Surface representation of the FAD-binding pocket in PqsL. Wild-type (green) and PqsL-Q (blue, PDB ID: 6SW2) FAD conformations are shown in stick representation. Pro273 (golden surface) directly interacts with the flavin, thereby occluding the flavin ring and hampering its interaction with NAD(P)H. **d** Interaction of Arg45 of PqsL-Q with the FAD pyrophosphate moiety. The conformation of Arg45 is further stabilized by hydrogen bonding to Glu106. **e** Comparison of the FAD conformations in PqsL wild type (green) and PqsL-Q (blue). The flavin of PqsL-Q is in a more outward position.

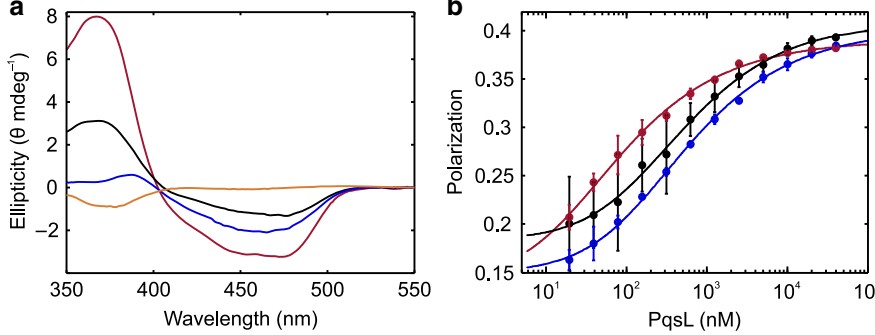

**Fig. 4 Spectroscopic assessment of flavin conformation and FAD-binding equilibria in PqsL. a** Circular dichroism spectra of FAD in solution (orange), PqsL (black), PqsL+2-ABA (blue) and PqsL-Q (red) (one representative measurement). **b** Dilution series of PqsL, monitored by FAD fluorescence polarization, to deduce $K_D$ of PqsL- (black), PqsL-2ABA- (blue), and PqsL-Q- (red) complexes with FAD (error bars indicate SD from three independent experiments). $K_D$ values were 249.5 ± 29.8 nM for the PqsL-, 259.8 ± 19.2 nM for PqsL-2ABA- and 37.5 ± 6.7 nM for the PqsL-Q-FAD complex.

conformation. Comparison with pHBH rather shows a local rearrangement of the substrate-binding site whose location with respect to the flavin site remains unchanged.

pHBH and similar enzymes have been shown to function through a fascinating molecular mechanism. The flavin oscillates between an IN position that promotes substrate oxygenation and an OUT position that is required for flavin reduction by NAD(P)H[13,20–24]. By comparison with the pHBH crystal structures, the flavin in PqsL can be described as in the OUT conformation (Fig. 3b). It should be noticed, however, that the flavin of PqsL is less accessible to solvent as compared to the OUT flavin of pHBH due a local conformational change that affects loop 272–276 (266–270 in pHBH). The shift of this loop, caused

by a kink at the Cα of Tyr276, positions Pro273 of PqsL (Pro267 in pHBH) in direct contact with the flavin, which thereby remains more buried within the protein compared to pHBH (Fig. 3c). In this conformation, there would be no space for the nicotinamide of NAD(P)H to bind close to the flavin in order to transfer its reducing hydride. An NAD(P)H-to-flavin hydride transfer reaction would require the flavin to move even further out of the active site or a conformational change of loop 272–276 to open the active site as observed in pHBH. We therefore tested whether manipulations in the area of the loop could render PqsL accessible to canonical light-independent reduction. To our surprise, the relatively drastic substitution of Pro273 by alanine led to stable protein that could be purified as described and

showed low catalytic activity in the PqsBC-coupled assay supported by FAD reductase HpaC. However, the variant exhibited only weak binding of FAD ($K_D = 64\,\mu M$) and, consequently, no detectable enzymatic oxidation of NAD(P)H. The direct interaction of Pro273 may thereby be a crucial factor for containing FAD within the catalytic pocket. It is therefore questionable whether tight flavin binding and NAD(P)H dependent hydride transfer are reconcilable within the PqsL architecture. Alternatively, the PqsL crystal structure would be fully compatible with flavin reduction through a photoinduced electron-transfer mechanism that does not involve direct hydride transfer and the physical overlap between the flavin and nicotinamide rings, as discussed in the next sections.

**Tight FAD binding does not interfere with photoreduction.** Practical challenges regarding PqsL are the enzyme's low tolerance for salt, heat, organic solvents, pH and buffer additives. In addition, PqsL shows reduced stability under anoxic conditions and has an intermediate affinity to FAD, leading to partial loss of cofactor in the process of purification. To render PqsL accessible to more demanding applications and analyses, we used a rational design approach, generating an enzyme variant with stronger FAD binding and thus improved stability.

Our starting point was the observation that PqsL displays a deviation in its FAD-binding site where Gly45 replaces an arginine side chain mostly conserved in pHBH-related enzymes. Specifically, the equivalent Arg44 of pHBH anchors the pyrophosphate group of the FAD. Moreover, this residue is part of a loop that is extensively involved in FAD- and NADPH-binding. Based on pHBH and PqsL crystal structures, we performed amino acid exchanges in PqsL aiming at the addition of functional groups in the vicinity of FAD ribityl and adenine nucleotide moieties for enhanced cofactor/cosubstrate coordination. In total, 34 protein variants were constructed and screened for in vivo activity using a recombinant *Pseudomonas putida* strain biotransformation assay[25] (Supplementary Table 2), the most promising of which was a quadruple mutant comprising the amino acid exchanges R41Y, I43R, G45R, C105G (henceforth termed PqsL-Q). Besides 72% activity in the *P. putida* screen, this variant outperformed the wild-type enzyme in terms of AQNO formed in *P. aeruginosa* (Supplementary Fig. 7). We also found that neither cosubstrate specificity (Supplementary Fig. 8), substrate affinity (Supplementary Fig. 6) nor $K_M$ values profoundly differ between PqsL and PqsL-Q (Supplementary Table 1), even though amino acid exchanges in PqsL-Q included positions designated to NADPH binding. The crystal structure of the PqsL-Q mutant revealed some local changes in the conformation of the 40-46 loop and confirmed that, as intended, Arg45 (replacing Gly45 of the wild-type) does engage the FAD pyrophosphate (Fig. 3d), which explains the 7-fold lower $K_D$ of FAD displayed by the mutant as compared to the wild-type protein ($37.5 \pm 6.7$ nM vs $249.5 \pm 29.8$ nM, respectively, Fig. 4b). The increased affinity for FAD likely explains another favorable feature of the mutant protein: purified PqsL-Q exhibited an increased thermostability and no degradation at room temperature (Supplementary Fig. 9). Moreover, the flavin of the mutant protein is slightly shifted outward with respect to the wild-type conformation (Fig. 3e).

However, the critical issue for this study was the extent to which PqsL-Q shared the photoswitchable catalytic activity of the wild-type enzyme. Quantification of the light-dependent activity of PqsL-Q using the PqsBC-coupled assay showed 20% product formation (Supplementary Fig. 10). The photoactivated reductive half reaction was assayed by NAD(P)H oxidation kinetics and revealed approximately 67% of wild-type activity. On the one hand, this almost unaltered activity provides yet another proof for

the protein-mediated electron transfer, because, due to the higher affinity of PqsL-Q to FAD, a reaction depending on residual free flavin would have been expected to be severalfold slower due to the reduced concentration of free FAD in the binding equilibrium. On the other hand, we can conclude that the more rigid conformation in which FAD is coordinated by PqsL-Q (see also Fig. 4a) apparently does not hinder the photocatalytic reaction.

**Semiquinone formation in PqsL photoreduction.** To unravel the mechanism of PqsL photoreduction, we illuminated PqsL-Q in a rapid mixing UV/Vis spectrophotometer under anoxic conditions with continuous high intensity blue light and recorded the induced spectral changes (Fig. 5a). Strikingly, the spectra show transient changes in the red-light region between 500 and 650 nm. The relatively slow emergence of a 575 nm species (Fig. 5b), as well as its general spectral characteristics are indicative of an intermittently formed neutral semiquinone. This intermediate is uncommon in flavoprotein monooxygenases in general and would imply that, rather than by a single-step transfer of a hydride ion, two sequential electron transfers would lead to the fully reduced flavin. The 450 nm transient shows a biphasic kinetic (Fig. 5c), supporting the notion that besides the oxidized FAD ($\varepsilon_{450\,nm} = 10.9$ mM$^{-1}$ cm$^{-1}$) and the fully reduced form ($\varepsilon_{450\,nm} = 1.8$ mM$^{-1}$ cm$^{-1}$), a third species emerges with medium absorptivity, which is consistent with the spectral properties of enzyme-bound neutral or anionic semiquinone[26].

To further confirm the existence of the semiquinone, we repeated the experiment in the presence of 1-hydroxy-3-methoxycarbonyl-2,2,5,5-tetramethylpyrrolidine (CMH), a cyclic hydroxylamine which is rapidly oxidized by free radicals to form a stable nitroxide radical (CM•). As seen in Fig. 5b, CMH lowers the total amount of semiquinone accumulated by increasing the apparent rate by which the radical is reduced. The spectroscopic indication of a single-electron transfer encouraged us to probe superoxide ($O_2^{•-}$) buildup when the reaction was carried out under aerobic conditions. We therefore monitored changes in oxygen consumption in steady state depending on the presence of superoxide dismutase (SOD, Fig. 5d). This resulted in a 19% decrease of oxygen consumption velocity, corresponding to the formation of at least 38% $O_2^{•-}$ during the reaction. Furthermore, when the radical scavengers SOD or CMH were added to the coupled assay, HQNO formation was increased, with a cumulative effect observed upon addition of both (Fig. 5e). This is consistent with two distinct effects: the very fast scavenging of harmful $O_2^{•-}$ arising from the reaction of NAD(P)(H)• with dioxygen[27], combined with the reduction of FAD•$^-$/FADH• to FADH$^-$/FADH$_2$ by CMH (which is capable of reducing the flavin semiquinone; NADPH is present in a vast excess, so that recovery of NADPH• by CMH would not influence reaction yields).

Multiple ways are conceivable by which a fully reduced flavin can form through photochemical and radical reactions with NADPH. With PqsL present in a dimeric topology under the assayed conditions, additional inter- and intradimeric charge transfers could occur, likely at different rates. Hence, we focused on the initial reaction of the photosensitized flavin, which constitutes the common starting point for all subsequent reaction routes. The simplest explanation for the observed spectral transitions would be a hydrogen atom transfer directly leading to the detected neutral semiquinone intermediate. To test this hypothesis, we probed the reaction with deuterated nicotinamide nucleotides. Charge transfer involving a deuteron would be accompanied by a reduction of the associated rate

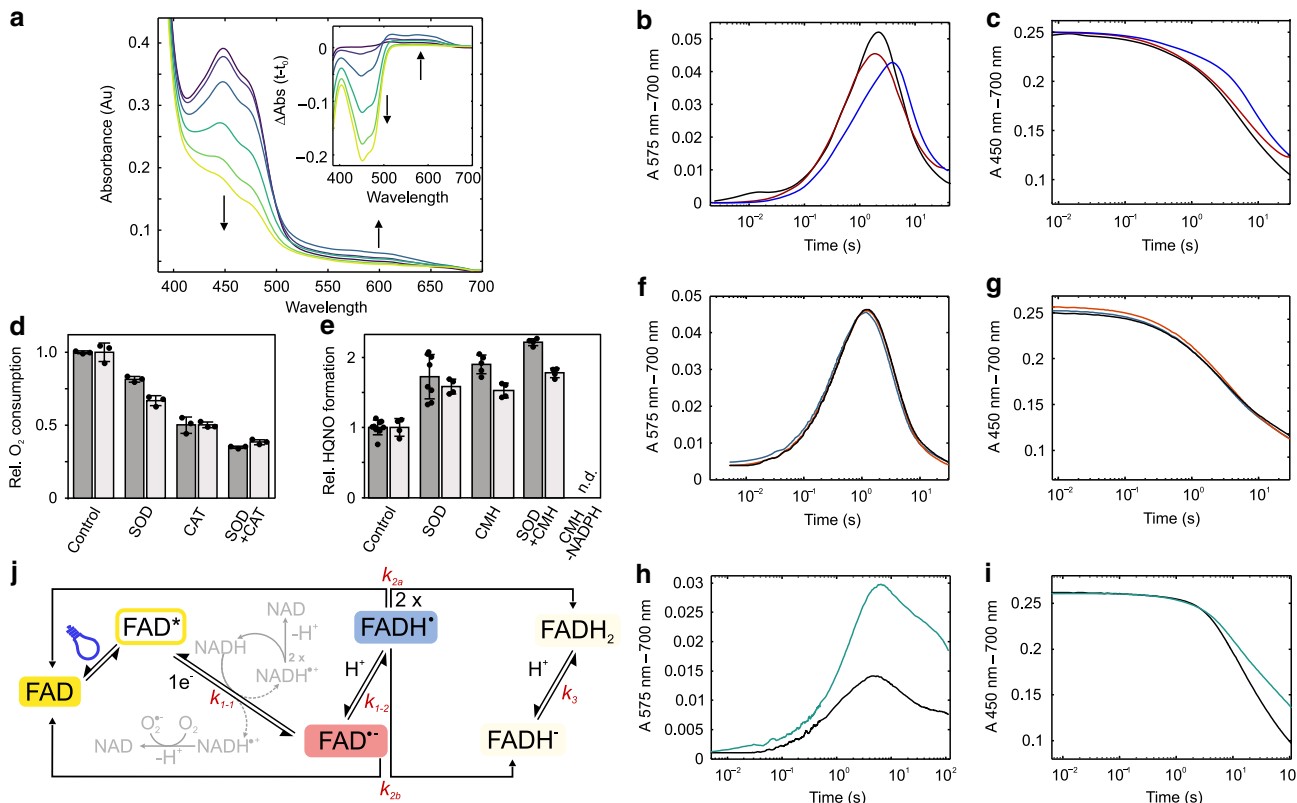

**Fig. 5 Flavin semiquinone formation in PqsL photoreduction. a** UV/Vis spectra and difference spectra (inset) at different time points (purple 0 s, violet 0.4 s, blue 0.8 s, dark green 2.4 s, light green 7.2 s, yellow 48 s) during photoreduction of PqsL-Q by continuous illumination under anoxic conditions in phosphate/borate buffer, pH 8.3 and 1 mM NADPH. **b, c** Rapid kinetics of PqsL-Q photoreduction with NADPH (black), NADPH+CMH (red), or NADH (blue) monitored at 575 nm (**b**) and 450 nm (**c**). Traces reflect averages of at least 10 replicates, data were corrected for baseline drift and scaled to the exact flavin content in the experiment. **d** Influence of $O_2^{\bullet-}$ scavenging superoxide dismutase (SOD), and/or $H_2O_2$ decomposing catalase (CAT) on oxygen consumption of PqsL (dark gray—PqsL, light gray—PqsL-Q, photoreduction with NADPH in a multiple turnover experiment). **e** Effect of SOD and/or the radical reducing CMH on HQNO formation in the coupled PqsL/PqsBC assay under constant illumination. Dark gray - PqsL, light gray - PqsL-Q. Error bars indicate SD of at least three independent measurements (panels **d**, **e**). **f, g** Rapid kinetics of PqsL-Q photoreduction with NADH (black), (4-R)-(4-²H)-NADH (orange) and (4-S)-(4-²H)-NADH (blue) at 575 nm (**f**) and 450 nm (**g**). **h, i** $D_2O$ solvent kinetic isotope effect on the photoreduction of PqsL-Q with NADH in tris-HCl/tris-DCl buffer at pH 8.35 (black) or pD 8.35 (green) monitored at 575 nm (**h**) and 450 nm (**i**). **j** Possible reaction routes of light-excited, PqsL-bound FAD. Bidirectional arrows reflect reactions in which the reverse rate could have significance. Side reactions of NADH (or NADPH) leading to superoxide formation are shown in gray.

constant caused by the higher molecular weight of the deuteron. Surprisingly, neither the *R*- nor the *S*-deuterated NADD enantiomer led to a kinetic isotope effect. Instead, the overall rate, as well as the apparent rates of accumulation and depletion of the neutral semiquinone remained remarkably consistent (Fig. 5f, g). This outcome eliminates hydrogen atom transfer as initial reaction step and also suggests that no protons originating from NADPH are translocated en route to fully reduced flavin.

We concluded that the transferred charge must be a single electron, resulting in the formation of a transient NADH•/FAD•− radical couple. The photoreaction of nicotinamides and flavins has been studied before and indeed suggests that, in solution, a single-electron transfer is the most probable intermediate of the reaction[28]. However, without spectroscopic evidence for the anionic semiquinone, we have to assume that initial electron transfer is followed by a very fast proton uptake that prevents even slight accumulation of the FAD•− species. To track flavin protonation, we compared reaction kinetics in water and deuterium oxide. Because PqsL shows strong precipitation in phosphate-buffered deuterium oxide, we substituted the buffer for tris-HCl/tris-DCl, which was avoided for other experiments due

to potential participation of tris in the flavin redox chemistry. As shown in Fig. 5h, i, deuterium oxide led to a stronger signal of the neutral semiquinone resulting from a slower decay of this intermediate. However, we could not detect any accumulation of a transient anionic semiquinone in $D_2O$, which indicates that both, protonation and deuteration are much faster than the initial electron transfer. Absorption transients at 575 nm suggest that $D_2O$ led to a slower decay of the neutral semiquinone, which was concomitant with a slower accumulation of the fully reduced flavin. No solvent kinetic isotope effects (SKIE) could be detected when the reaction was performed with free flavin under the same conditions, confirming that the observed kinetic difference is characteristic of the protein-associated semiquinone. We calculated a $D_2O/H_2O$ SKIE of the semiquinone decay in the order of 50 (details below), which is extremely high, but not unprecedented in enzymatic reactions[29]. To explain this outcome, several aspects must be taken into account. The formation of fully reduced flavin can occur in several reactions, each of which involves the disproportionation of a radical couple. Due to the very short lifetime of the NADH• radical of only a few microseconds[30] we can assume that flavin interaction with NADH• plays a subordinate role whereas the disproportionation

of two flavin semiquinones has a much higher incidence. In addition, with low abundance of the anionic semiquinone, the involvement of this species must also be relatively limited, leaving the dismutation of two neutral semiquinones as the major contributor to generate fully reduced flavin. In solution, semiquinone disproportionation is very fast and difficult to observe during the photochemical reduction[31]. Hence, PqsL must exert a stabilizing effect on the semiquinone, e.g., through spatial shielding and molecular interaction. Due to the altered characteristics of deuterium bonds, which, depending on donors and acceptors involved, can exhibit higher binding energies[32], it is conceivable that this shielding effect is enhanced in $D_2O$, thereby disfavoring dismutation. It is to be expected that primary kinetic isotope effects caused directly by the proton-coupled flavin-flavin charge transfer have the most impact on the observed kinetic shift. Evidence exists that SKIEs are particularly severe when donor-acceptor distances are large[33]. It is conceivable that this criterion applies to the PqsL–PqsL FAD radical dismutation.

In addition, it is unclear to what extent protein interactions play a role in semiquinone disproportionation. Because PqsL is a non-obligatory dimer that monomerizes at high NaCl concentrations (see Supplementary Fig. 11), it is possible that rearrangements within the protein population may influence semiquinone reactivity. The higher viscosity of $D_2O$ and potentially enhanced dimer stability due to higher deuterium bond energy could therefore slow down disproportionation. Remarkably, as shown in Supplementary Fig. 12, the productivity of PqsL, at least in the range covered in this assay, is not concentration-dependent, which argues against any cooperative effects or diffusion as major factor of the reaction. Nevertheless, the actual formation of $FADH_2/FADD_2$, albeit probably a fast reaction, also involves a proton/deuteron rearrangement, which must be impacted by a kinetic isotope effect. Eventually, it is known that deuterated solvents generally reduce spin–spin relaxation and therefore lead to increased lifetime of flavin semiquinones[34]. The sum of these contributions can explain the substantially longer lifetime of the deuterated semiquinone.

The overall reaction can be modeled according to the reaction scheme shown in Fig. 5j, using available extinction coefficients for the neutral semiquinone at 450 and 575 nm[35]. A global fit comprising two sequential reactions suggests that photoreduction and protonation can be described with an apparent pseudo first order rate. Under the conditions used, we observe a 30% faster rate for NADPH than for NADH ($k_1 = 0.6$ s$^{-1}$ for NADPH and $k_1 = 0.4$ s$^{-1}$ for NADH). This discrepancy can explain the higher productivity with NADPH (see Fig. 1c) and higher oxygen consumption in steady state (see Fig. 2d). Subsequent semiquinone disproportionation can be described by a second order reaction. This rate is affected by the buffer, with tris leading to faster decay, which could be explained by, e.g., increased formation of free flavin radicals formed by direct reduction with tris. These species potentially support disproportionation by reacting with the protein-bound semiquinone. In addition, tris may also influence flavin coordination by PqsL. The apparent rates determined in the global fit are $k_2$ ($H_2O$) $= 1.7 \times 10^6$ M$^{-1}$ s$^{-1}$ and $k_2$ ($D_2O$) $= 3.3 \times 10^4$ M$^{-1}$ s$^{-1}$.

PqsL instability precludes some experiments that would have been helpful to further characterize semiquinone dismutation. For example, variation of pH/pD could reveal whether FAD$^{•-}$ protonation was unspecific, or depended on a protein-associated donor, such as a side chain or a coordinated water. In addition, fast EPR spectroscopy could be used to verify the existence of both the anionic semiquinone species, as well as the NADH radical, and reveal specifics of the protein-bound photosensitized flavin. Nevertheless, our experiments provide evidence for a photoinduced, protein-mediated radical mechanism of NAD(P)

H-flavin electron transfer that may serve as example for the development of photoactivatable flavoenzymes.

**Substrate-independency of PqsL photoreduction.** Our study reports on the protein-mediated photoreduction of a flavoprotein monooxygenase. It moreover demonstrates an enzymatic radical reaction of the nicotinamide-flavin redox couple, a reaction that could be of greater relevance than anticipated so far, not only for biocatalytic applications but also as a critical side reaction in various physiological contexts. Based on structural and spectroscopic studies, we hypothesize that prerequisites of photoreactivity are the binding of both, FAD and NAD(P)H, a permanent OUT conformation of the flavin, and steric impairment of a direct nicotinamide-flavin stacking to avoid photoindependent nicotinamide oxidation by the canonical hydride transfer mechanism.

Even though 2-ABA, the substrate of PqsL, shows an influence on the photoreaction, the reaction is essentially independent of the substrate. It is therefore feasible that the scheme by which PqsL photoreacts with NAD(P)H may be transferable to a multitude of related enzymes, either to enforce NAD(P)H-dependent activity in enzymes naturally requiring pre-reduced flavin, or to substitute the reductive half reaction, establishing artificial photoactivity for use in, e.g., cascading biocatalytic processes. First candidates could be enzymes with a glutathione reductase II fold and no nicotinamide-dependent activity, such as two-component monooxygenases (Class E) or certain halogenases (formally class F monooxygenases)[13].

Another promising perspective is the construction of photoenzymes with covalently bound flavin cofactor (e.g., based on the halogenase ClmS[36]). In contrast to reactions harnessing the photochemistry of free flavin, enzyme-bound cofactors are protected so that intramolecular photochemical side reactions are suppressed. A covalently bound flavin would therefore constitute the ideal setting for a robust, long-lasting flavin photocatalyst.

## Methods

**Plasmids, protein production, and purification**. Heterologous production of PqsL and its variants was performed using *E. coli* LMG194 (Thermo Fisher Scientific) harboring the pBAD-based *Escherichia-Pseudomonas* shuttle vector pHERD30T[37] containing the respective genes. Mutations in *pqsL* were introduced by site-directed mutagenesis using the primers listed in Supplementary Table 3. If necessary, nucleotides were exchanged by sequential PCRs. For main cultures, 2YT medium containing 20 µg ml$^{-1}$ gentamicin were inoculated to OD$_{600}$ of 0.05 and cultivated at 37 °C, 130 rpm until an OD$_{600}$ of 0.5 was reached. Cultures were cooled down to 20 °C for 40 min, subsequently induced with 0.02% (w/v) L-arabinose and incubated overnight. Cells were harvested by centrifugation (8000 × *g*, 15 min, 4 °C) and resuspended in lysis buffer (20 mM tris-HCl, 150 mM NaCl, 5 mM imidazole, 0.1% (v/v) Nonidet P-40, 10% (v/v) glycerol, pH 8.5) prior to sonication (5 × 4 min, 50% amplitude, 50% duty cycle, Hielscher UP200S). Cell lysates were centrifuged (42,000 × *g*, 1 h, 4 °C) and supernatants filtered through a 0.45 µM PVDF syringe filter (Carl Roth). PqsL was purified by immobilized nickel affinity chromatography using a 5 ml Ni-NTA agarose column (Macherey-Nagel, Düren, Germany) and ÄKTA FPLC (GE Healthcare) for automated chromatography. After application of the soluble protein extract, the column was washed with 20 column volumes of washing buffer (20 mM tris-HCl, 300 mM NaCl, 10 mM imidazole, 10% (v/v) glycerol, pH 8.5) and proteins were eluted with 20 mM tris-HCl, 300 mM NaCl, 150 mM imidazole, 5% (v/v) glycerol, pH 8.5). After dialysis against 20 mM tris-HCl, 1 mM DTT, 10% (v/v) glycerol, pH 8.5 overnight at 4 °C, PqsL was applied to a 5 ml Resource Q anion exchange chromatography column (GE Healthcare). The column was washed with 10 column volumes of 20 mM tris-HCl, pH 8 before a gradient elution (0–500 mM NaCl, 20 mM tris-HCl, pH 8 over 20 column volumes) was applied. PqsL was concentrated to approximately 500 µM with a 10 kDa MWCO centrifugal concentrator (Sartorius), aliquoted and frozen in liquid nitrogen. Purity of PqsL was analyzed by SDS-PAGE using a discontinuous 12.5% gel (Supplementary Fig. 13) Production and purification of PqsBC and HpaC was performed as described previously[38,39].

**Synthesis of deuterated nicotinamide adenine dinucleotides**. The (4-*R*)-(4-²H) and (4-*S*)-(4-²H) stereoisomers of reduced nicotinamide adenine dinucleotide

(NADD) were prepared based on procedures described by Viola et al., 1978[40]. Ethanol-$d_6$, glucose-1-$d$, *Saccharomyces cerevisiae* alcohol dehydrogenase, *S. cerevisiae* aldehyde dehydrogenase and *Leuconostoc mesenteroides* glucose-6-phosphate dehydrogenase were purchased from Merck KGaA (Darmstadt, Germany). Synthesized NADD was purified by anion exchange chromatography using a 6 mL Resource 15Q column (GE Healthcare). To this end, the complete reaction mixture was loaded onto the column equilibrated with buffer A (20 mM tris-HCl, pH 7.7) and eluted with a gradient to 40% buffer B (20 mM tris-HCl, 500 mM KCl, pH 7.7) over 60 ml at 1 ml min$^{-1}$ flow. NADD eluted at approximately 8.5% buffer B. Purity was monitored by $A_{260}$ to $A_{340}$ absorbance ratio for all collected fractions. Fractions with a $A_{260}/A_{340}$ of ≤2.3, indicating fully reduced NADD, were pooled, assayed for deuterium content by LC/MS (Supplementary Fig. 14) and lyophilized.

**Analytical gel filtration.** Analytical gel filtration chromatography of PqsL was performed using a Superdex 200 Increase 10/300 GL equilibrated with either 20 mM tris-HCl, pH 8.5 or 20 mM tris-HCl, pH 8.5, 150 mM NaCl at a constant flow rate of 0.6 ml min$^{-1}$ at 12 °C. For the estimation of molecular weight, a calibration run for both buffer conditions with a Bio-Rad gel filtration standard (#1511901) was performed.

**Analysis of PqsL in *Pseudomonas putida* and *P. aeruginosa*.** The in vivo activity of PqsL and its variants was indirectly measured by analysis of ratios between the alkyl quinolones (AQ) 2-heptyl-4(1*H*)-quinolone (HHQ), 2-heptyl-3-hydroxy-4 (1*H*)-quinolone (PQS) and 2-heptyl-4-hydroxyquinoline-*N*-oxide (HQNO) in bacterial cultures. For this purpose, *P. putida* harboring *pqsABCD*[25] and thus capable of 2-ABA, the central intermediate of AQ biosynthesis, and HHQ synthesis was transformed with the respective *pqsL*-containing expression plasmids (Supplementary Table 2) by electroporation. Alternatively, *P. aeruginosa ΔpqsL* was scarlessly complemented with *pqsL* or *pqsL-Q* as described previously[41]. Afterwards, main cultures containing the respective antibiotics were inoculated to an OD$_{600}$ of 0.05 and incubated for 16 h at 30 °C (*P. putida*) or 37 °C (*P. aeruginosa*) and 160 rpm. Alkyl quinolones were extracted from bacterial cultures by addition of 1 volume of acidic ethyl acetate (0.1% (v/v) acetic acid) and vigorous vortexing. Organic phases from two repeated extraction steps were collected and dried *in vacuo*. Metabolites were analyzed by HPLC.

**Determination of binding affinities.** Affinities of PqsL to FAD or 2-ABA were determined by fluorescence spectroscopic techniques[38]. Binding assays based on serial dilutions of PqsL in 20 mM tris-HCl, pH 8.5 were performed by fluorescence polarization spectrometry (excitation at 450 nm, 20 nm bandwidth; emission at 520 nm, 20 nm bandwidth) using a FP-6500 spectrofluorometer (Jasco) with polarization accessory. Dissociation constants of PqsL–FAD complexes were calculated by non-linear regression using equation

$$FP = FP_0 + (FP_{max} - FP_0)\left(\frac{K_D + (\frac{1}{occ} + 1)[FAD] - \sqrt{(K_D + (\frac{1}{occ} + 1)[FAD])^2 - \frac{4[FAD]^2}{occ}}}{\frac{2[FAD]}{occ}}\right)$$

(1)

where FP is measured fluorescence polarization, $FP_0$ is FP of free FAD, $FP_{max}$ is theoretical maximum of FAD–protein complex, $K_D$ is dissociation constant of the FAD–protein complex, [FAD] is concentration of FAD–protein complex (varied) and occ is FAD occupancy of the FAD–protein complex in percent.

**Analysis of light-induced steady-state oxidation activity.** Oxygen consumption of the light-driven cosubstrate oxidation by PqsL was measured using a Digital 10 Clark electrode (Rank Brothers Ltd.). The 300 μl reaction mixture was composed of either 10 μM PqsL or 10 μM free FAD and varying concentrations of an electron donor, e. g. NADPH, NADH, 1-methyl-1,4-dihydronicotinamide, ascorbic acid, EDTA, L-cysteine, L-glycine and triethylamine (7.8–10,000 μM) in 30 mM phosphate/borate, pH 8.3. If indicated, 2-ABA was added to a final concentration of 200 μM. After 30 s of equilibration in the stirred oxygen electrode reaction chamber at room temperature, illumination with a high-performance blue-light LED ($λ_{max}$ = 466 nm, 20 nm full width at half maximum (FWHM), 1050 μmol photons s$^{-1}$ m$^{-2}$) was applied and oxygen consumption was followed.

To study the ratio between superoxide and hydrogen peroxide released during the reaction, 10 μM PqsL with 1 mM NADPH were illuminated in the presence of either or both, superoxide dismutase (20 U ml$^{-1}$) or catalase (200 U ml$^{-1}$). The amount of reactive oxygen species produced was then estimated from the differences in apparent oxygen consumption.

For analysis of the wavelength dependency of the light-driven reduction of PqsL by NADPH, the reaction setup was coupled to the light source of a Jasco FP-6500 spectrofluorometer equipped with a fiberoptic accessory for illumination of the reaction chamber. Photon counts for each wavelength were measured with a TIDAS diode array detector (J&M Analytik, Essingen, Germany), so that reaction rates could be normalized accordingly.

**Light-dependent product formation.** For estimation of the light-dependent hydroxylation of 2-ABA, 100 μl of a reaction mixture containing 10 μM PqsL, 1 mM electron donor (NADPH, NADH, 1-methyl-1,4-dihydronicotinamide, ascorbic acid, EDTA, L-cysteine, L-glycine, triethylamine), 100 μM 2-ABA, 120 μM octanoyl-CoA and 100 U catalase in 20 mM tris-HCl, pH 8.3 was illuminated ($λ_{max}$ = 466 nm, 20 nm FWHM, 1050 μmol photons s$^{-1}$ m$^{-2}$) for 10 min. If indicated, 500 μM freshly prepared 1-hydroxy-3-methoxycarbonyl-2,2,5,5-tetramethylpyrrolidine (CMH) was included in the mix. For control reactions, 0.2 μM of the NADH-dependent flavin reductase HpaC was added. In these cases, only NADH was used as electron source and the samples were not illuminated. Afterwards, PqsBC was added to a final concentration of 0.5 μM and illumination was continued for 10 min. If indicated, light intensity was modified by stepwise increase of the distance of the LEDs in 1 cm increments. Intensities were determined for each increment using a LI-COR Model-Li189 Quantum Photometer. The reaction was stopped by addition of 200 μL ethyl acetate (0.1% acetic acid). Resulting alkyl quinolones 2-heptyl-4(1 *H*)-quinolone (HHQ) and 2-heptyl-4-hydroxyquinoline-*N*-oxide (HQNO) were then extracted by liquid-liquid extraction using ethyl acetate and quantified via HPLC.

**Light-dependent NAD(P)H oxidation.** Light-driven NAD(P)H oxidation was studied by illuminating 100 μl reaction mixture (10 μM PqsL, 1 mM NAD(P)H in 20 mM tris-HCl, pH 8.3) with blue light ($λ_{max}$ = 466 nm, 1050 μmol photons s$^{-1}$ m$^{-2}$) for the indicated intervals. Afterwards, the reaction products were directly analyzed by HPLC.

**Photodegradation of FAD.** Degradation of FAD in free and PqsL-complexed states was analyzed by illuminating solutions containing 500 μM free FAD or PqsL in 30 mM tris-HCl, pH 8.3 with a blue LED. Samples were taken after 0, 10, 30, and 60 min of illumination, diluted 1:10 in 15% (v/v) acetonitrile buffered with 0.1% (v/v) formic acid, centrifuged for 1 min at 20,000 x *g*. The supernatant was directly subjected to HPLC. Decomposition of FAD was followed by monitoring the formation of lumichrome, the major degradation product of FAD at pH 8.3[4].

**HPLC and LC-MS analysis.** All HPLC analyses were performed with an Agilent 1100 Series instrument (Agilent, Santa Clara, CA, USA) equipped with an Eurospher II 3 μm C18 100 × 3 mm column (Knauer, Berlin, Germany).

Analysis of alkyl quinolones was carried out within 6 min at 0.6 ml min$^{-1}$ isocratic flow in 70% methanol/water buffered with 0.1% (v/v) acetic acid. Separation of NAD$^+$, NADP$^+$, NADH and NADPH was achieved by 4 min of isocratic flow of 15 mM ammonium acetate, pH 6.8, followed by a linear gradient from 0 to 15% methanol over 10 min at a flow rate of 0.25 ml min$^{-1}$. FAD and its derivatives were separated by 6 min of isocratic flow in 15% acetonitrile/water buffered with 0.1% (v/v) formic acid, ensued by a linear gradient from 15 to 53% acetonitrile over 24 min and 0.5 ml min$^{-1}$ flow.

HPLC-MS analyses of (deuterated) reduced nicotinamide adenine dinucleotides were performed using a Thermo Fisher Scientific Ultimate 3000/ Bruker Amazon speed instrument (Bruker Daltonics, Billerica, MA, USA) equipped with a Knauer Eurospher II 100-5 C18 150 × 3 mm column. Solvents were 15 mM aqueous ammonium acetate, pH 6.8 (A) and acetonitrile (B). Elution was started with 5 min of isocratic flow (100% A) before linearly increasing to 20% B over 30 min. Afterwards, the column was washed for 5 min (90% B) and equilibrated to starting conditions for another 5 min. Flow was kept constant at 0.5 ml min$^{-1}$. Electrospray ionization parameters were set according to the manufacturer's recommendations.

**Rapid kinetics of the PqsL reaction.** All reactions were conducted in oxygen-free buffer (30 mM phosphate/borate, pH 8.3) at 4 °C using an SFM-3 stopped-flow instrument (BioLogic, Seyssinet-Pariset, France) coupled to a TIDAS (J&M Analytik, Essingen, Germany) diode array detector and operated with the proprietary software. The instrument was made anoxic by incubating the syringes and reaction chambers with an anoxic oxygen purging buffer solution (10 mM glucose, 100 μg ml$^{-1}$ glucose oxidase, 5 μg ml$^{-1}$ catalase, 20 mM MOPS, pH 6.5) overnight. The flow system was rinsed prior to the experiments with anoxic sample buffer. To analyze the light-induced reductive half reaction, PqsL was rapidly mixed with NADH, NADPH, (4-*R*)-NADD or (4-*S*)-NADD, respectively, to final concentrations of 15 μM PqsL and 1 mM reduced nicotinamide adenine dinucleotide. For analyses of solvent kinetic isotope effects, all phosphate/borate buffers were exchanged for 30 mM tris-DCl in D$_2$O pD 8.35 (equivalent to pH = 8.17). Prior to the experiments, 300 μL of concentrated PqsL was dialyzed against 15 ml 30 mM tris-DCl in D$_2$O pD 8.35 (pH = 8.17) for 2 h followed by dialysis overnight in fresh buffer at 8 °C. Experiments were conducted at 6 °C to avoid freezing of D$_2$O solutions in the reservoirs of the stopped-flow instrument. Reference experiments were conducted in 30 mM tris-HCl buffers at pH 8.35. The cuvette was illuminated in a 90° angle to the optic path using a blue LED ($λ_{max}$ = 466 nm), whereas the white light source (150 W xenon lamp, 2 mm aperture) used for spectroscopy was attenuated to 10% strength with a neutral-density filter for minimal interference with the recorded kinetics. Reference measurements in the absence of illumination were recorded for correction of artifacts caused by the low intensity light used for spectroscopic monitoring. Data were processed using wavelet denoising and 2D penalized least squares smoothing[42]. Individual measurements of each reaction were averaged

(n > 5) and corrected by scaled subtraction of the respective $t-t_0$ dark difference spectra. Reaction kinetics were analyzed and fit according to different models using Matlab R2018b or Dynafit 4.

**Crystallographic studies**. Purified enzymes used for crystallization experiments were concentrated up to 10 mg ml$^{-1}$ in 20 mM tris-HCl, pH 8.0, 150 mM NaCl, 5% v/v glycerol. We performed initial crystallization screening experiments using wild-type PqsL with and without incubation with its substrate 2-ABA (400 µM) using an Oryx 8 robot (Douglas Instruments, UK). Experiments were performed in sitting drop using MRC SWISS CI plates, with a 1:1 protein:reservoir ratio and 0.2–1 µl crystallization droplets. Crystals grew in several conditions with best results from JCSG CORE II kit in a condition containing PEG 20,000 (5–10% w/v) and MES buffer, pH 6.5. These conditions were then used for all (wild-type and mutant) crystallization experiments. Crystals harvested from mother liquor using nylon cryo-loops (Hampton Research/Molecular Dimensions) were shortly soaked in a cryoprotectant solution containing 15–20% v/v PEG 400, 5% w/v PEG 20,000, 0.1 M sodium chloride, 0.1 M MES, pH 6.5, 2 mM 2-ABA and then flash-frozen in liquid nitrogen. Diffraction data were measured on the beam-lines of the ESRF Grenoble (France). Data processing was performed with XDS[43] and programs of the CCP4 suite[44]. The structure of the wild-type PqsL was solved by molecular replacement using the PDB 2X3N structure as search model and programs of the CCP4. Electron density map was then inspected, and structure refinement was performed with COOT[45], REFMAC[46] and other programs from CCP4 suite. The substrate 2-ABA could be clearly identified in the active site of the enzyme as described in the results. Despite extensive attempts, no structure with bound NADP(H) could be obtained. The structure of the PqsL-Q quadruple mutant (R41Y, I43R, G45R, C105G) was solved using the same procedures and the wild-type PqsL structure as initial model. The crystallographic statistics are shown in Table 1.

**CD spectroscopy**. CD spectra of PqsL were recorded with a Jasco J-600 spectro-meter at room temperature, using a 10 mm quartz cuvette. Samples contained either PqsL, PqsL-Q or FAD (10 µM) and 100 µM 2-ABA (if indicated) in 20 mM tris-HCl buffer, pH 8.3. For data collection, four scans (100 nm s$^{-1}$, 2 nm band-width) were recoded and ellipticities were calculated within the Jasco spectra manager software. Spectra were subsequently denoised by wavelet filtering and averaged in Matlab R2018b.

**Reporting summary**. Further information on research design is available in the Nature Research Reporting Summary linked to this article.

## Data availability
Further data supporting this study are available in the Supplementary Information. The datasets generated and analyzed during the current study are available from the corresponding author on reasonable request. Coordinates and structure factors have been deposited in the Protein Data Bank under accession codes 6SW1 and 6SW2.

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

## Acknowledgements
The authors thank U. Hennecke for provision of 2-nitrobenzoylacetate, A. Jagels and H.-U. Humpf for access to CD spectrometer, the European Synchrotron Radiation Facility (ESRF) and the Swiss Light Source (SLS) for the provision of synchrotron radiation facilities and beamline scientists of the ESRF and the SLS for assistance. This work was supported by the Deutsche Forschungsgemeinschaft, grant FE 383/23-2, the Italian Ministry of Education, University and Research (MIUR): Dipartimenti di Eccellenza Program (2018–2022) – Department of Biology and Biotechnology "L. Spallanzani", University of Pavia and by the European Union's Horizon 2020 research and innovation program under the Marie Skłodowska-Curie grant agreement No 722390.

## Author contributions
S.E. and S.D. designed and performed experiments, S.E., S.D., and S.F. analyzed data, S.R. and A.M. performed crystallizations and solved and analyzed X-ray structures, S.D., S.E., A.M., and S.F. wrote the paper.

## Competing interests
The authors declare no competing interests.
