## [Peer Review File · Nature Communications]

Reviewers' comments:

Reviewer #1 (Remarks to the Author):

The authors have suitably answered my comments. Therefore, I have no objections to publish this manuscript.

Reviewer #2 (Remarks to the Author):

This paper is a reviewed version of a manuscript that describes unanticipated light dependent NADPH reduction of flavin. Nicotinamide mediated flavin reduction is a very common reaction normally understood as a hydride transfer step, while the fact light can influence or even drive enzyme activity is something that is becoming a hot topic (be it artificial or natural).

As such, this manuscript provides interesting angle: an apparently standard flavoenzyme requires blue light to be reduced by NADPH. The authors spend considerable effort in characterising this in detail and propose a single electron transfer mechanism.

To what extent this applies in vivo remains to be established, as well as the finer details of the mechanism here. However, the revised version addressed the detailed comments previously made by the three reviewers in a very constructive manner and the main conclusions are solidly supported by the data.

I recommend publication.

Reviewer #3 (Remarks to the Author):

Overall, the authors have adequately addressed most of the issues that I mentioned in my review. However, I would like to have clarification on the following points before recommending publication.

Page 6, Line 156: I am still having difficulty understanding the relative rates of the enzymatic and the non-enzymatic reactions. I would expect that the relative rates would be largest when the enzyme is saturated. This does not seem to be the case. The authors make their measurement at $[\text{NADPH}] = 7.8 \mu\text{M}$ but the K_m for NADPH is $158 \mu\text{M}$. Please explain.

Figure 2, Legend, Line 131: I assume that the experiment in panel D (dotted lines) is for the non-enzymatic reaction and uses $[\text{FAD}] = [\text{PqsL}]$. If so, that is an important control experiment. However, it demonstrated that the enzyme has only a very small effect on the reaction rate. This needs clarification.

Page 14, line 382: A solvent deuterium isotope effect of 50 is unprecedented (as far as I know) in enzyme-catalyzed reactions and merits discussion. How does this number compare with the isotope effect on the non-enzymatic reaction?

Reviewers' comments:

Reviewer #1 (Remarks to the Author):

The authors have suitably answered my comments. Therefore, I have no objections to publish this manuscript.

Reviewer #2 (Remarks to the Author):

This paper is a reviewed version of a manuscript that describes unanticipated light dependent NADPH reduction of flavin. Nicotinamide mediated flavin reduction is a very common reaction normally understood as a hydride transfer step, while the fact light can influence or even drive enzyme activity is something that is becoming a hot topic (be it artificial or natural).

As such, this manuscript provides interesting angle: an apparently standard flavoenzyme requires blue light to be reduced by NADPH. The authors spend considerable effort in characterising this in detail and propose a single electron transfer mechanism.

To what extent this applies in vivo remains to be established, as well as the finer details of the mechanism here. However, the revised version addressed the detailed comments previously made by the three reviewers in a very constructive manner and the main conclusions are solidly supported by the data.

I recommend publication.

Reviewer #3 (Remarks to the Author):

Overall, the authors have adequately addressed most of the issues that I mentioned in my review. However, I would like to have clarification on the following points before recommending publication.

Page 6, Line 156: I am still having difficulty understanding the relative rates of the enzymatic and the non-enzymatic reactions. I would expect that the relative rates would be largest when the enzyme is saturated. This does not seem to be the case. The authors make their measurement at $[\text{NADPH}] = 7.8 \mu\text{M}$ but the K_m for NADPH is $158 \mu\text{M}$. Please explain.

The fact that the relative rate advantage (i.e. ratio of $k_{\text{cat}}(\text{PqsL}) / k(\text{FAD/NAD(P)H photoreduction})$) of the enzymatic reaction is highest at a relatively low NAD(P)H concentration is very characteristic when comparing an enzymatic and a non-enzymatic reaction which both proceed at similar rates (here k_{cat} vs. the pseudo-first order rate at a given FAD concentration). This is because the non-enzymatic reaction, $-\text{d}[\text{O}_2] = k * [\text{FAD}] * [\text{NAD(P)H}]$, is linearly correlated to $[\text{NAD(P)H}]$, whereas the enzymatic reaction is amplified at low concentrations due to the formation of the Michaelis-Menten complex. Increasing $[\text{NAD(P)H}]$ will result in linear convergence of the non-enzymatic reaction toward its maximum rate (see Fig. 2d, orange trace/triangles), which may eventually catch up with the enzymatic reaction. The enzymatic reaction will remain more or less constant after reaching 90 % of k_{cat} already at $[\text{NADPH}] = 5 * K_m$. Therefore, at higher concentrations of electron donor, the ratio of cat./uncat. reactions becomes smaller.

We decided to mention the 30-fold ratio at $7.8 \mu\text{M}$ NADPH, because it nicely illustrates a major advantage of an enzyme-associated photoreduction, which is its high efficiency at low concentrations of electron donor available. We modified the manuscript text to address this aspect in more detail (line 140 ff).

Figure 2, Legend, Line 131: I assume that the experiment in panel D (dotted lines) is for the non-enzymatic reaction and uses $[\text{FAD}] = [\text{PqsL}]$. If so, that is an important control experiment. However, it demonstrated that the enzyme has only a very small effect on the reaction rate. This needs clarification.

No, this is a misunderstanding. The non-enzymatic reaction is only reflected by the orange trace/solid triangles.

The dotted traces/open symbols reflect rates of the purified, FAD-saturated enzyme (i.e. purified, highly concentrated PqsL, supplemented with excess FAD, then passed through a desalting column). These rates contain a contribution free FAD, which is released from the complex due to the dilution of PqsL in the sample (calculated by the K_D of the PqsL-FAD complex). The solid traces are corrected for this contribution and contain the rates for the actual PqsL-FAD holoenzyme. The corrected traces run higher, because the enzymatic reaction is faster than the non-catalyzed reaction. We think the dotted traces are useful because they reflect the actual, measured data without further correction (except for the normalization), whereas the solid traces are more accurate from a biochemical point of view. We revised the figure legend for improved clarity as follows: “...Kinetics of light-induced oxygen consumption by PqsL supplemented with NADPH (blue), NADH (violet), NADPH + 2-ABA (green) and NADH + 2-ABA (ocher) or free FAD + NADPH (orange). Rates of the PqsL reactions were corrected for the contribution of residual free FAD (calculated from K_D of the PqsL-FAD complex) and FAD photodegradation. Dashed lines/open circles show the rates of FAD-saturated PqsL prior to any corrections (i.e. rates as measured, normalized to FAD content).”

Page 14, line 382: A solvent deuterium isotope effect of 50 is unprecedented (as far as I know) in enzyme-catalyzed reactions and merits discussion.

We appreciate the reviewer's suggestion to provide a more detailed discussion of this experiment. We decided to make a first mention of the SKIE earlier in the paragraph (line 305) and added additional aspects that may further explain its extent in lines 318 ff. The observed SKIE indeed is uncommonly strong and probably unprecedented in flavin chemistry. However, very large KIEs and SKIEs are observed in enzymatic and non-enzymatic reactions and also in reactions involving a proton-coupled charge transfer (e.g., SKIE of 8 for hydroperoxyl deprotonation: *Arch Biochem Biophys.* 2011 Mar 1; 507(1): 36–43, SKIEs of up to 35 in Ruthenium-bipyridine complex proton coupled electron transfer: *J. Am. Chem. Soc.* 2001, 123, 16, 3723-3733, KIE of 5.7 in the flavin reduction of D-amino acid oxidase: *Biochemistry* 1992, 31, 8207-8215, KIE of 60 in lipoxigenase: *J. Am. Chem. Soc.* 1994, 116, 2, 793-794).

We would like to mention that we assayed solvent influence on the photoreduction of free flavin as well. Using the same conditions PqsL was tested, we did not detect any measurable SKIE. We added a comment on this outcome to the manuscript text in line 302 f. We are also not aware of any study focusing on the SKIEs of flavin in photochemical reactions in water, probably because semiquinone disproportionation in freely moving flavins is so much faster than photoreduction, hence difficult to investigate and probably not particularly interesting.

How does this number compare with the isotope effect on the non-enzymatic reaction?

Flavin undergoes numerous side reactions independent of the designated electron donor which would have to be taken in account for a calculation of the net nicotinamide-flavin electron transfer rates. Hence, to present a photoreduction kinetic for FAD or any other flavin compound these side reactions would have to be analyzed individually and tested for their dependency on the particular electron donor employed. Therefore, we refrain from a more detailed comparison of the catalyzed and non-catalyzed reactions. This would be a project on its own, beyond the scope of this study.

REVIEWERS' COMMENTS:

Reviewer #3 (Remarks to the Author):

The authors have adequately addressed my questions and the paper is now suitable for publication.